# Vapor Compounds Released from Nicotine-Free Inhalators as a Smoking-Cessation Aid

**Ho-Seok Kwak [1] , Jung-Yeol Han [2], Gideon Koren [3], Sang-Hee Jo [4] and Ki-Hyun Kim [5],***

[1]  Department of Bioanalytical Mass Spectrometry, Sunin Bio Corporation, Seoul 07528, Korea; duck-ac@hanmail.net
[2]  Department of Obstetrics and Gynecology, National Medical Center, Seoul 04564, Korea; hanjungyeol055@nmc.or.kr
[3]  Motherisk Israel and Maccabi–Kahn Institute of Research and Innovation, Tel Aviv 6812509, Israel; koren_gid@mac.org.il
[4]  Research Division for Industry & Environment, Korea Atomic Energy Research Institute, Jeongeup-si, Jeollabuk-do 56212, Korea; shjo@kaeri.re.kr
[5]  Department of Civil & Environmental Engineering, Hanyang University, Seoul 04763, Korea
*  Correspondence: kkim61@hanyang.ac.kr; Tel.: +82-2-2220-2325; Fax: +82-2-2220-0399

**Abstract:** The health risks of cigarette smoking have been reported to increase continuously, while it is estimated to be responsible for the death toll of more than seven million people globally each year. In an effort to reduce the risk involved in cigarette smoking, nicotine-free inhalators have been developed as smoking-cessation aids. To evaluate the feasibility of nicotine-free inhalators in such respect, we investigated the composition of volatile organic compounds (VOCs) released from the consumption of nicotine-free inhalators of which major components include natural essential oils and traditional Chinese medicinal herbs. Vapor samples from nicotine-free inhalators were generated and collected for analysis using an e-cigarette auto-sampler. The vapor samples were captured onto a multi-bed sorbent tube sampler and a 2,4-dinitrophenylhydrazine (DNPH) cartridge for the quantitative analysis with the aid of thermal desorption-gas chromatography/mass spectrometry and high-performance liquid chromatography, respectively. A total of 29 VOCs were determined in vapor samples at concentrations below 0.2 ppm. Concentrations of (+)-isomenthone and acrolein slightly exceeded the derived no-effect level (DNEL) or sensory irritation level. However, VOCs were below the concentration exposure limit, according to the Occupational Safety and Health Administration (OSHA). According to our study, most of the aroma compounds and VOCs released from nicotine-free inhalator were lower than the DNEL or sensory irritation level. Consequently, it is found that nicotine-free inhalators could be safe to use in reference to toxic guidelines for inhalation exposure. However, if the use of nicotine-free inhalators is over prolonged period, it can also increase the risk of exposure to potentially toxic compounds.

**Keywords:** aroma essential oil; cigarette smoking; smoking-cessation aid; inhalator; no smoking

---

## 1. Introduction

The cigarette-related death toll is estimated at around seven million people annually [1]. Hence, the cessation of cigarette-smoking has become one of the critical components in establishing public health policies or associated regulations [2]. Many smoking-cessation tools and devices (e.g., nicotine patch, nicotine inhalator, and nicotine nasal spray) have been developed and introduced into the market; nicotine-free inhalators with aroma essential oils are one particular product available commercially in Korea and some other countries [3–8]. It was reported that nicotine-free inhalators were free from the addictive effect of nicotine [8].

The nicotine-free inhalator was developed to offer psychological comfort to curb many smoking-associated habits through the simulation of smoking conventional cigarettes [3,9]. The nicotine-free inhalator may be beneficial for use in smoking-cessation interventions, particularly in smokers for whom handling and manipulating their cigarettes is an important part of the smoking ritual (e.g., the hand-to-mouth action of smoking) [8,10]. Therefore, nicotine-free inhalators are expected to be one of the effective options to make smokers refrain from their smoking habit.

A typical nicotine-free inhalator device consists of an air-breathing system that does not require electronic operation [5,6,8]. It is simply a fibrous, sponge-filter plug soaked in naturally extracted herbal oil, which is encased in a plastic cartridge container similarly to a cigarette [8]. The nicotine-free inhalator is different from other cigarette-related products (e.g., traditional cigarettes, e-cigarettes, and nicotine products) in that it does not contain typical cigarette chemical compounds (e.g., nicotine, propylene glycol, or vegetable glycerin). The herbal materials that are infused into the nicotine-free inhalator are made from natural essential oils and herbs [8,11]. The nicotine-free inhalator can be purchased online [5,6]. Nonetheless, there have not been sufficient efforts to evaluate the human health impacts of volatile organic compounds (VOCs) released from nicotine-free inhalators. Therefore, many questions arise regarding the safety of these products when used for therapeutic treatment. For this reason, we aim to evaluate the safety and risks associated with VOCs released from nicotine-free inhalators when used by consumers. Herein, we analyzed for the first time the volatile compounds of vapor from marketed nicotine-free inhalators.

## 2. Materials and Methods

### 2.1. Collection of Vapor Samples Released from Nicotine-Free Inhalators

A nicotine-free inhalator (aroma non-smoking pipe; Mihyang Med. Inc., Seoul, Korea) is a smoking-cessation device that does not require electric power for operation. It is composed of a sponge-filter and herbal materials that are encased in a plastic cartridge container like an electronic cigarette device. An e-cigarette auto-sampler designed by Chemtekins (Gyeonggi-do, Korea) was used for the generation and collection of e-cigarette aerosol samples. The e-cigarette auto-sampler consists of five main parts: a pneumatic control valve, a flow controller, an e-cigarette holder, a six-port valve, and a console. Sampling was carried out following the procedures for e-cigarette smoke analysis recommended by the National Institute of Food and Drug Safety Evaluation (NIFDS) in Korea as follows [12]: Vapor samples were regularly generated using an e-cigarette auto-sampler (Chemtekins, Korea) under a fixed inhalation condition (1 L/min of inhalation flow rate, 2 s of inhalation time, 10 s interval between inhalations, resulting in 330 mL of total sampling volume) [13,14]. After a nicotine-free inhalator was placed in the e-cigarette auto-sampler, vapor samples were captured onto each of two sampling filters: (1) a multi-bed sorbent tube filled with Tenax TA and Sulficarb (Markes International, Llantrisant, UK) to collect the VOCs, and (2) a 2,4-dinitrophenylhydrazine (DNPH) cartridge (TOP Trading ENG Co., LTD, Seoul, Korea) to collect the carbonyl compounds. The pretreatment and calibration methods for samples collected onto the two filters are described in Sections 2.2 and 2.3. The concentrations of VOCs and carbonyl compounds in vapor samples were measured from duplicate samples collected from the nicotine-free inhalator products.

### 2.2. Analysis of Carbonyl Compounds in Vapor Samples

Generation or detection of carbonyl compounds, especially for formaldehyde, in electronic cigarette vapor is regarded as significant human health threat. Thus, the amount of 10 target carbonyl compounds (formaldehyde, acetaldehyde, propionaldehyde, butyraldehyde, isovaleraldehyde, valeraldehyde, acrolein, acetone, crotonaldehyde, and benzaldehyde) in the vapor samples from the nicotine-free inhalator were evaluated using high-performance liquid chromatography (HPLC) equipped with a UV detector system (Spectra system UV2000, Thermo Scientific, Walthan, MA, USA). For carbonyl-compound analyses, an HPLC system was used with a mixture containing 15 μg/mL each

of 10 carbonyl-DNPH derivatives for the standard (Supelco, Bellefonte, PA, USA). The standards were diluted with 99.9% acetonitrile (J. T. Baker, Phillipsburg, NJ, USA) so that standard solutions could be prepared for calibration analysis at five different concentrations (0.15, 0.30, 0.75, 1.50, and 3.00 ng/μL).

The carbonyl components in the vapor samples collected via DNPH derivatization were eluted with 5 mL of acetonitrile. In addition, blank concentrations of carbonyl compounds detected in the DNPH cartridges were also measured (e.g., below method detection limit (MDL) values). Each sample elution was then injected into the inlet of the HPLC system through a 20-μL sample loop and then separated on an Acclaim 120 C18 column (particle size: 5 μm, diameter: 4.6 mm, length: 250 mm; Dionex, Sunnyvale, CA, USA) using a mobile phase of acetonitrile:water (7:3) at a fixed flow rate of 1.5 mL/min. Finally, the carbonyl-hydrazones were detected by a UV detector (Dionex, Sunnyvale, CA, USA) at a wavelength of 360 nm. This analytical method was adopted for the determination of carbonyls in vapor samples in our previous research [15].

### 2.3. Analysis of Volatile Organic Compounds in Vapor Samples

The vapor samples collected onto a multi-bed sorbent tube were analyzed with thermal desorption-gas chromatography (TD-GC)/mass spectrometry (MS) system (TD: Unity-xr, Markes International; GC: GC-2010 Plus, Shimadzu, Kyoto, Japan; MS: GCMS-QP2020, Shimadzu, Kyoto, Japan) to quantify 19 volatile compounds. Among them, 19 compounds (methyl ethyl ketone, methyl isobutyl ketone, butyl acetate, isobutyl alcohol, benzene, toluene, p-xylene, m-xylene, o-xylene, styrene, dimethyl disulfide, trimethylamine, propionic acid, butyric acid, isobutyric acid, hexanoic acid, heptanoic acid, phenol, and m-cresol) were quantified using standard materials. In order to prepare standard solutions for the 19 compounds, standards were purchased as liquids (Sigma-Aldrich, Louis, MO, USA; all standards were 97–99.5% pure, except for Trimethylamine (25.0%)). Solutions were prepared by diluting each standard in 100% methanol (J.T. Baker) to cover a range of concentration levels (5, 10, 20, 50, and 100 ng/μL). Then, for calibration analyses, 1 μL of each standard solution was spiked into the sorbent tube using a 5 μL syringe (SGE Analytical Science, Victoria, Australia), while supplying pure (99.999%) $N_2$ gas at 50 mL/min for 4 min using a mini pump (MP-Σ30, SIBATA, Tokyo, Japan). Each sorbent tube containing volatile organic compounds in vapor samples and standard solutions was then placed in the thermal desorption (TD) and thermally vaporized at 320 °C for 10 min. These vaporized gas samples were preconcentrated in a cold trap (Carbopack C and Carbopack B combination) at −25 °C and then thermally desorbed at 320 °C for 10 min. A total of 25 compounds were separated on a DB-Wax column (film thickness: 0.25 μm, diameter: 0.25 mm, length: 60 m; Agilent J&W, Santa Clara, CA, USA) for detection by GC/MS. The MS system was operated in electron ionization (EI) mode at 70 eV. Ions were monitored in full-scan mode (range: 35–400 m/z).

### 2.4. Quantification Method Based on the Carbon Number Approach Without Standards

Twelve CLASS compounds (compounds lacking authentic standards or surrogates) were quantified without using standards (e.g., α-pinene, β-pinene, β-myrcene, limonene, 3-octanol, (+)-isomenthone, (−)-isomenthone, menthyl acetate, neomenthol, menthol, piperitone, and m-eugenol). Although standards for these compounds are not available, their presences were detectable in the form of large peaks in the chromatograms obtained by the vapor sample analyses. These 12 CLASS species were thus quantified based on the effective carbon number theory [16,17], which is defined in relation to the sum of the carbon number in each compound's molecular structure. First, "predicted response factor (RF) values" were assigned to each of the 25 standard compounds; predicted RF is based on the theoretical RF value but quantified using the total mass spectrum (35–400 m/z) applied in this experiment. Then, an empirical equation was developed to predict the RF values of the 25 (CLASS) VOCs based on the relationship between carbon numbers of 25 standards and their RF values (Figure 1). The RF values for 12 CLASS compounds were thus estimated using the relationship derived from 25 standard compounds. For example, the RF value of α-pinene was predicted as 108,8703 by putting its carbon number (10) into the relationship.

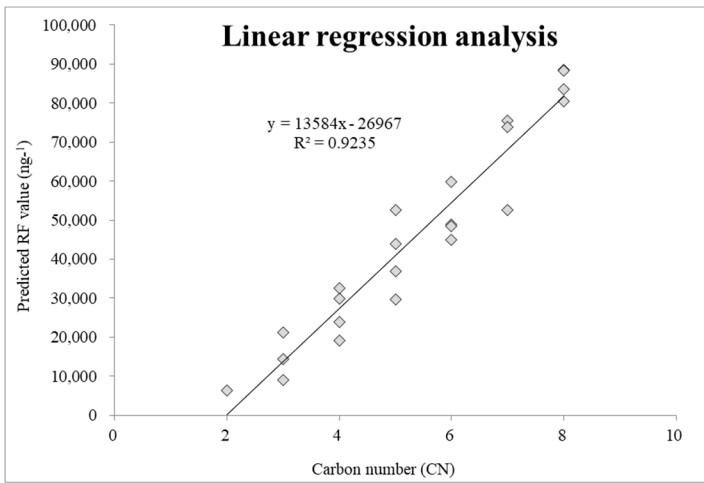

**Figure 1.** Plot of response factor (RF) values against carbon number (CN) of 25 standard compounds. The equation derived from these standard species was used to predict RF values of CLASS (compounds lacking authentic standards or surrogates) volatile organic compounds (VOCs).

## 3. Results

### 3.1. Method Validation of GC-MS and HPLC

In this study, the reliability of our experimental procedures was validated as part of basic quality assurance (QA) procedures [12–17]. In HPLC analysis, the calibration curves (CCs) showed excellent linearity ($R^2 > 0.99$) for all (n = 10) target CCs. The relative standard deviation (RSD) values were also calculated from triplicate analyses of the standard solution (0.75 ng/μL). The resulting RSD values of targets ranged from 1.77 to 7.64%. The method detection limit (MDL), if assessed in terms of absolute mass (ng), ranged from 0.10 to 0.65 ng. In GC-MS analysis, the target VOCs showed linearity within acceptance criteria ($R^2 > 0.99$). The resulting RSD values ranged from 0.44 to 8.04%. The MDL of VOCs ranged from 0.07 to 4.72 ng.

### 3.2. Concentration of VOCs and Aroma Compounds/Data of Toxic Guidelines

Table 1 summarizes the aroma compounds collected as vapor samples from the nicotine-free inhalator. A total of 12 aroma compounds were identified through sorbent tube sampling of volatiles based on GC/MS analyses.

The concentrations of these compounds ranged from 0.26–7.09 ppm. The most abundant vapor compounds were menthol (5.49 ppm), (±)-isomenthone (3.84–7.09 ppm), and limonene (3.39 ppm). Other minor constituents were also found, including neomenthol (1.32 ppm), 3-octanol (1.28 ppm), α-pinene (0.73 ppm), β-pinene (0.90 ppm), β-myrcene (0.74 ppm), menthyl acetate (0.74 ppm), piperitone (0.26 ppm), m-eugenol (0.25 ppm). All detected compounds in vapor samples were identified as peppermint oil components [18,19].

Permissible inhalation exposure limits were compared with our sampled data to assess the toxic effects of these compounds. Permissible exposure limits were based on guidelines from the Occupational Safety and Health Administration (OSHA), the derived no-effect level (DNEL), and sensory irritation levels. Menthol concentration was higher than the sensory irritation level but lower than DNEL. Concentration of (+)-isomenthone was more than DNEL (6.35 ppm), but sensory irritation values for (+)-isomenthone were not available. Myrcene and piperitone have not been identified as inhalation toxins by the $RD_{50}$ (exposure concentration corresponding to a 50% respiratory rate decrease) and DNEL [20,21]. Furthermore, other compounds were observed in lower concentrations than sensory irritation and DNEL levels.

Figure 2 displays chemical structures of aroma compounds detected from vapor samples from a nicotine-free inhalator, which contained alcohols (1 compound: 3-octanol) and monoterpenes

(11 compounds). The most widely present class of hydrocarbons in essential oils was identified as the terpenes. The basic component of diverse terpene compounds is a five-carbon molecule called an isoprene. As a chemical bond forms between two isoprene units, a 10-carbon molecule called a monoterpene can be produced. Monoterpenes are widely distributed in plants and show volatility during use of essential oils [22].

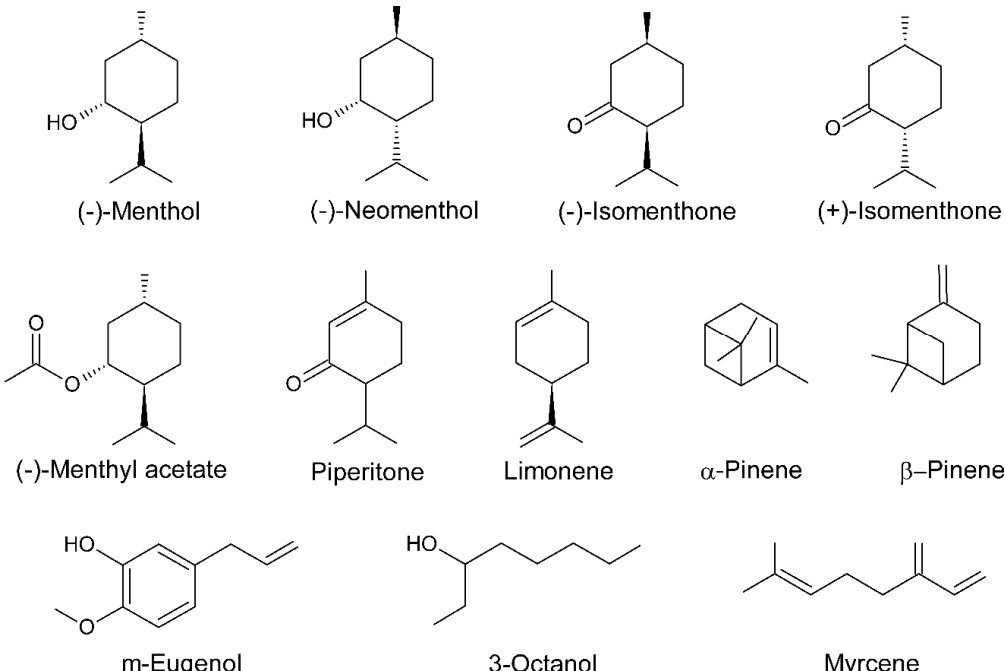

**Figure 2.** Chemical structure of volatile aroma compounds in vapor samples from a nicotine-free inhalator.

**Table 1.** Concentrations of aroma compounds quantified in vapor samples taken from a nicotine-free inhalator and toxic guidelines.

| No | Compound | Chemical Abstracts Service (CAS) Number | Vapor Samples (n = 4) Mean Conc. (ppm) | RD$_{50}$ [1] (ppm) | Sensory Irritation [2] (ppm) | DNEL [3] (ppm) |
|---|---|---|---|---|---|---|
| 1 | (+)-α-Pinene | 7785-70-8 | 0.73 ± 0.19 | 1053 | 31.6 | 0.69 |
| 2 | (−)-β-Pinene | 18172-67-3 | 0.90 ± 0.19 | 4663 | 140 | 1.04 |
| 3 | β-Myrcene | 123-35-3 | n.d. [4] | n.d. [4] | - | n.d. |
| 4 | (+)-Limonene | 138-86-3 | 3.39 ± 0.47 | 1076 | 32.3 | 12.14 |
| 5 | 3-Octanol | 589-98-0 | 1.28 ± 0.14 | 256 | 7.68 | n.d. |
| 6 | (+)-Isomenthone | 491-07-6 | 7.09 ± 0.84 | n.d. | - | 6.35 |
| 7 | (−)-Isomenthone | 1196-31-2 | 3.84 ± 0.31 | n.d. | - | 6.35 |
| 8 | Menthyl acetate | 89-48-5 | 0.74 ± 0.04 | n.d. | - | 4.20 |
| 9 | Neomenthol | 491-01-0 | 1.32 ± 0.08 | n.d. | - | 10.49 |
| 10 | (−)-Menthol | 1490-04-6 | 5.49 ± 0.31 | 45 | 1.35 | 20.95 |
| 11 | Piperitone | 89-81-6 | 0.26 ± 0.01 | n.d. | - | n.d. |
| 12 | m-Eugenol | 501-19-9 | 0.25 ± 0.04 | n.d. | - | 3.20 |

[1] RD$_{50}$: concentration that depresses the respiratory rate by 50% due to sensory irritation [23]; [2] Sensory irritation = 0.03 × RD$_{50}$ [23]; [3] DNEL: derived no-effect level for substances assumed to have a threshold exposure level [24]; [4] n.d.: no data.

Table 2 summarizes the VOCs present in vapor samples of a nicotine-free inhalator. A total of 29 VOCs were identified by the HPLC-UV and GC/MS methods. Concentrations of all VOCs were estimated to be <0.2 ppm. Aldehydes (nine compounds) and acetone measured by the HPLC-UV method were quantified and ranged from 0.05 ppm (propionaldehyde) to 0.11 ppm (benzaldehyde). A total of 19 VOCs measured by the GC/MS method were quantified as present up to 0.06 ppm. VOCs detected above 0.001 ppm (or 1 ppb) were observed in the following order: toluene (0.0609 ppm), methyl ethyl ketone (0.0280 ppm), total xylene (0.0162 ppm), styrene (0.0030 ppm), and hexanoic

acid (0.0016 ppm). The other 12 VOCs were quantified as present at a level lower than 1 ppb. Concentrations of all VOCs were lower than the exposure limit (sensory irritation, DNEL, and OSHA). The concentration of acrolein was above the sensory irritation level but lower than the DNEL and OSHA criteria.

**Table 2.** Concentrations of volatile organic compounds (VOCs) in vapor samples from a nicotine-free inhalator and toxic guidelines as a reference.

| No | Compound | CAS Number | Vapor Samples (n = 4) Mean Conc. (ppm) | $RD_{50}$ [1] (ppm) | Sensory Irritation [2] (ppm) | DNEL [3] (ppm) | OSHA PEL TWA [4] (ppm) |
|----|----------|-----------|------------------------|----------|----------|------|------|
| 1 | Formaldehyde | 50-00-0 | 0.06 [5] | 4 | 0.12 | 7.43 | 0.75 |
| 2 | Acetaldehyde | 75-07-0 | 0.08 [5] | 2845 | 85.4 | n.d. | 200 |
| 3 | Propionaldehyde | 123-38-6 | 0.05 [5] | 2078 | 62.3 | 2.60 | n.d. |
| 4 | Butyraldehyde | 123-72-8 | 0.08 [5] | 1015 | 30.5 | n.d. | n.d. |
| 5 | Isovaleraldehyde | 590-86-3 | 0.07 [5] | 757 | 22.7 | n.d. | n.d. |
| 6 | Valeraldehyde | 110-62-3 | 0.10 [5] | 1121 | 33.6 | n.d. | n.d. |
| 7 | Acrolein | 107-02-8 | 0.06 [5] | 1.6 | 0.05 | 0.09 | 0.1 |
| 8 | Acetone | 67-64-1 | 0.06 [5] | 23,480 | 704 | 517 | 1000 |
| 9 | Crotonaldehyde | 123-73-9 | 0.08 [5] | 3.53 | 0.10 | 0.11 | 2 |
| 10 | Benzaldehyde | 100-52-7 | 0.11 [5] | 333 | 10.0 | 2.29 | n.d. |
| 11 | Methyl ethyl ketone | 78-93-3 | 0.0280 ± 0.0068 | 9000 | 270 | 206 | 200 |
| 12 | Methyl isobutyl ketone | 108-10-1 | 0.0001 [5] | 3195 | 95.9 | 20.6 | 100 |
| 13 | Butyl acetate | 123-86-4 | 0.0001 [5] | 730 | 21.9 | 64.1 | 150 |
| 14 | Isobutyl alcohol | 78-83-1 | 0.0001 [5] | 1818 | 54.5 | n.d. | 100 |
| 15 | Benzene | 71-43-2 | 0.0012 [5] | n.d. | - | n.d. | 1 |
| 16 | Toluene | 108-88-3 | 0.0609 ± 0.0192 | 3373 | 101 | 51.7 | 200 |
| 17 | p-Xylene | 106-42-3 | 0.0043 ± 0.0011 | 1326 | 39.8 | 51.6 | 100 |
| 18 | m-Xylene | 108-38-3 | 0.0031 ± 0.0010 | 1360 | 40.8 | 51.6 | 100 |
| 19 | o-Xylene | 95-47-6 | 0.0088 ± 0.0060 | 1467 | 44.0 | 51.6 | 100 |
| 20 | Styrene | 100-42-5 | 0.003 [5] | 156.3 | 4.70 | 23.8 | 100 |
| 21 | Dimethyl disulfide | 624-92-0 | 0.0002 [5] | n.d. | - | 0.57 | n.d. |
| 22 | Trimethylamine | 75-50-3 | 0.0003 [5] | 61 | 1.83 | 2.06 | n.d. |
| 23 | Propionic acid | 79-09-4 | 0.0005 [5] | 386 | 11.6 | 24.4 | n.d. |
| 24 | n-Butyric acid | 107-92-6 | 0.0003 [5] | 285 | 8.55 | 10.4 | n.d. |
| 25 | i-Butyric acid | 79-31-2 | 0.0001 [5] | n.d. | - | 51.8 | n.d. |
| 26 | Hexanoic acid | 142-62-1 | 0.0016 ± 0.0002 | n.d. | - | 3.76 | n.d. |
| 27 | Heptanoic acid | 111-14-8 | 0.0001 [5] | n.d. | - | 4.03 | n.d. |
| 28 | Phenol | 108-95-2 | 0.0005 [5] | 166 | 4.98 | 2.11 | 5 |
| 29 | m-Cresol | 108-39-4 | 0.0006 ± 0.0002 | n.d. | - | 0.80 | n.d. |

[1] $RD_{50}$: concentration that depresses the respiratory rate by 50% due to sensory irritation; [2] Sensory irritation = $0.03 \times RD_{50}$; [3] DNEL: derived no-effect level for substances assumed to have a threshold exposure level; [4] OSHA PEL: the permissible exposure limit (PEL) of hazardous substances according to the occupational safety and health administration (OSHA), TWA: 8 hour time-weighted average; [5] This value was estimated to be below the method detection limit (MDL).

## 4. Discussion

### 4.1. Volatile Compounds of the Nicotine-Free Inhalator

The composition of mint oils was found to be stable based on regular GC analysis [25]. In another study, a total of 41 compounds were identified in the peppermint (*Mentha piperita* L.) essential oil by headspace/solid phase microextraction (HS/SPME)-GC/MS and hydrodistillation-GC/MS [18]. The main components that were present at concentrations higher than 1% in the hydrodistillation sampling method were menthol (45.34%), menthone (16.04%), menthofuran (8.91%), cis-carane (8.70%), 1,8-cineole (4.46%), neomenthol (4.24%), and limonene (2.22%). The main components that were present at concentrations higher than 1% in the HS/SPME method were menthol (29.38%), menthone (16.88%), cis-carane (14.39%), menthofuran (11.38%), 1,8-cineole (9.45%), trans-caryophyllene (2.76%), neomenthol (2.37%), β-pinene (2.26%), α-pinene (1.55%), germacrene-D (1.41%), trans-sabinene hydrate (1.28%), and neoisomenthyl acetate (1.02%). The HS/SPME sampling method resulted in relatively higher amounts of high-volatile monoterpenes and lower detection of less volatile compounds such as menthol and menthone, compared with solvent-based samples from essential oil distillation [18].

All compounds detected in the nicotine-free inhalator are generally comparable to those commonly found in peppermint oil. Some peppermint oil components (e.g., cis-carane, menthofuran,

and 1,8-cineole) were not detected in the vapor sample from the nicotine-free inhalator. It appears that some ingredients were removed during the manufacturing process. All VOCs, with the exception of acrolein, were estimated to be present at lower concentrations than the sensory irritation concentration. The increased concentration of acrolein is thought to be associated with herb processing, because when glycerol is heated, it decomposes into acrolein. Consequently, a small amount of glycerol in herbs is likely to be converted into acrolein during the manufacturing process [26].

### 4.2. Toxicity Guidelines for Inhalation Exposure

Sensory airway irritation is a sign that a chemical substance is stimulating the nerves of the respiratory tract as identified by a characteristic pause during expiration on a respiratory sensory irritation test [27]. Sensory irritation is commonly used as a parameter for setting occupational exposure limits. The no-observed-adverse-effect level (NOAEL) should be estimated from $RD_{50}$ value found in animals [23]. $RD_{50}$ values should be used in NOAEL estimation for comparison from human studies [28].

The $RD_{50}$ is a useful guide for setting protective levels for the health of both workers and general users who can be exposed to a target substance. Kuwabara et al. reported a strong correlation between $RD_{50}$s and a number of reference values, such as the lowest observed adverse effect levels (LOAELs), threshold limit values (TLVs), and California reference exposure levels (RELs) [29]. The relationship between $RD_{50}$ and these other threshold guidelines could be used to identify protective values for the public to prevent respiratory or sensory irritation [29]. An examination of TLVs and $RD_{50}$ values showed a high correlation between TLVs and $0.03 \times RD_{50}$, which supports the continued use of animal bioassays for establishing exposure limits to prevent sensory irritation in the workplace [23].

In the REACH (Registration, Evaluation, Authorisation and Restriction of Chemicals)-registered substances database [24], DNEL values can be found for different exposure routes (e.g., oral, inhalation, and dermal), for different population types (e.g., workers or consumers), and for different exposure frequencies (e.g., short- or long-term). DNEL values are therefore useful for determining inhalative occupational exposure or long-term inhalative consumer exposure under the REACH regulations [30].

The OSHA, a U.S. regulatory agency established in 1970, incorporated these values as a reference for permissible exposure limits (PELs) to 102 VOCs of industrial importance. The time-weighted average (TWA) concentrations for OSHA PELs must not be exceeded during any 8 h work shift of a 40 h work week [31]. An unsuccessful attempt was made to determine the inhalation $RD_{50}$ for benzene [27]. The investigators found that the respiratory rate of mice increased after 5 min of exposure to 5800 ppm (18,800 mg/m$^3$) benzene.

### 4.3. Safety of the Nicotine-Free Inhalator

Peppermint oil is widely used in folk remedies and traditional medicine to treat digestive disorders and nervous system problems because of its antitumor, chemopreventive, and antimicrobial properties, as well as its beneficial renal and antiallergenic effects [32–34]. It is also thought to be effective against cramping, digestive complaints, anorexia, nausea, and diarrhea [32,33]. Peppermint oil is composed of menthol and menthone, along with several other minor constituents [32]. Although most peppermint research has been conducted on rat models, it is considered a medicinal plant for human diseases. However, peppermint oil has also been found to cause cyst-like changes in the white matter of the cerebellum and nephropathy at overdoses of 40–100 mg/kg per day for 28–90 days [32]. In rats, 80 mg and 160 mg pulegone doses over 28 days caused atonia, weight loss, decreased blood creatinine, and histopathological changes in the liver and the white matter of the cerebellum. Menthol was also observed to cause hepatocellular changes in rats [32]. During the Cosmetic Ingredient Review (CIR) Expert Panel Meeting, the panel expressed concern over oral dosing of peppermint oil, as some studies found evidence of pulegone-based toxicity [35].

The pulegone and menthofuran in peppermint oil are hepatotoxic; thus, they should be restricted in food and herbal products. The CIR commission should consider the latest risk assessments on pulegone and menthofuran: according to a public statement from the European Medicines Agency on

the use of herbal medicinal products, *M. piperita* (peppermint) oil contains a maximum of 4% pulegone and between 1% and 9% menthofuran [35]. Furthermore, the Scientific Committee on Food (SCF) has concluded that pulegone is mainly metabolized through pathways involving menthofuran, and these two substances show similar toxicity. The authors of "Essential Oil Safety" have recommended a maximum daily oral dose of 152 mg for *M. piperita* oil [36]. This oral restriction is based on an 8% menthofuran and 3% pulegone content, with limits of 0.2 mg/kg/day for menthofuran and 0.5 mg/kg/day for pulegone [35].

Toxic substances (menthofuran and pulegone) have been detected in vapor analyses of peppermint oil [18], but they were not detected in our nicotine-free inhalator. Levels of aroma substances and VOCs that are detected at lower levels than DNEL or sensory irritation values can be considered as safe. However, the isomenthone and acrolein levels were detected to be higher than those of the DNEL or sensory irritation values that are considered safe (depending on total use time). Acrolein is considered to pose one of the non-cancerous health risk of all hazardous air pollutants [37]. In animal studies, decreased body weight gains were reported in rats, hamsters, monkeys, and rabbits exposed to 0.32–4.9 ppm acrolein. In the intermediate exposure study in rats (61 days, 24 h/day), the NOAEL of 0.06 ppm and LOAEL of 0.32 ppm have been calculated [38]. The acrolein level (0.06 ppm) in the nicotine-free inhalator is the same as NOAEL for intermediate exposure. If daily exposure time is clearly shorter than 24 h, it is considered safe.

The usage pattern of a nicotine-free inhalator is typically intermittent, such as when the consumer feels a smoking craving. The typical usage time of a nicotine-free inhalator is usually much shorter than a worker's exposure time (8 h). However, if the usage time of a nicotine-free inhalator exceeds eight hours per day, its exposure risk may go beyond guideline levels.

## 5. Conclusions

This study was carried out to assess the safety of a nicotine-free inhalator in reference to toxic guidelines for inhalation exposure. As such, it is likely to represent the very first attempt to evaluate its safety with respect to the major and minor components involved in such system. The findings of this study indicate that nicotine-free inhalators can be reliably used as a safe smoking-cessation aid. However, the scope of our study is limited by the fact that our data were not derived from a human exposure study. Thus, further research is needed to confirm the amount of human exposure in relation to usage patterns from nicotine-free inhalator use. Nevertheless, this study presents the possibility that nicotine-free inhalators can be used safely by smokers who want to stop smoking.

This study confirmed that most commonly used aroma compounds and VOCs are detectable and quantifiable at low levels relative to DNEL or sensory irritation values and likely safe. However, if the use of a nicotine-free inhalator is over a prolonged period, its exposure could be harmful.

**Author Contributions:** Conceptualization, methodology, validation, and formal analysis were done by H.-S.K., J.-Y.H., S.-H.J., G.K., and K.-H.K. Further, investigation, writing, editing, and all other miscellaneous tasks were done by K.-H.K. and J.-Y.H.

**Funding:** This research provided by the Ministry of Environment (Grant No: 2018001850001) as well as by a grant from the National Research Foundation of Korea (NRF), which is funded by the Ministry of Science, ICT & Future Planning (grant No: 2016R1E1A1A01940995).

**Acknowledgments:** The authors thank the Ministry of Environment and the National Research Foundation of Korea for financial support.

**Conflicts of Interest:** All authors declare that there is no conflict of interest.

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
