# Peer review of "Vapor Compounds Released from Nicotine-Free Inhalators as a Smoking-Cessation Aid"

_applsci, doi:10.3390/app9112232_

Round 1
Reviewer 1 Report
Summary:
This study is centered around a commercially available nicotine-free inhaler manufactured commercially by Mihyang Med, Inc. The authors use a commercially available e-cigarette autosampler to simulate puffs off the inhaler, and collect each puff on two filters: a filter with Tenax TA and Sulficarb sorbents for VOC analysis in GC/MS, and a filter with DNPH for the analysis of carbonyls in an HPLC. The study seems to be well thought through, however it is poorly written. Details of the setup are skipped, no chromatograms are available, and connection between the results and safe concentrations of inhaled compounds are not apparent. I have a few suggestions, offered below for the authors’ consideration, after which I think this manuscript will be of higher quality and appropriate for Applied Sciences. I do intend to raise an important query: the authors declare no conflicts of interest, yet to my understanding the study was carried out on an inhaler manufactured by Mihyang Med, Inc., who funded part of the study. Both the Abstract and the Conclusion sections claim there is no health risk if used<8h/day, but some compounds are above DNEL. This needs to be addressed clearly.
Major Comments:
Abstract: No major comments; some grammatical errors but on the whole very concise and informative!
1. Introduction: Overall, the introduction is very short and the English needs to be revisited. I am not sure if the authors do not have access to enough journals, but I would suggest a more in-depth review of the literature. How many types of these smoking-cessation devices that are not e-cigarettes or nicotine products exist (L35 the authors claim ‘many’, they can perhaps elaborate)? Are they mostly commercial or are there any being developed by research institutions? How does aerosolization work in these vs. e-cigarettes? Is the only difference the absence of nicotine, nicotyrene, PG, or PEG? Have there been studies that show these devices are effective? If most of this information is proprietary, perhaps the authors can clarify that.
2. Materials and Methods: Sections 2.3 and 2.4 are fairly well described (however figures of calibrations or chromatograms are desired); but Sections 2.1 and 2.2 are too vague. It is my understanding that the nicotine-free inhaler (inhalator?) was connected to an autosampler, and the ~330 mL sample was sent to a filter, then the filter to the the HPLC. This isn’t clear, though. The authors need to go into more detail on their methods, be, and explain why they chose to detect carbonyls amongst other VOCs. Also, I do not understand how the sample on the filter gets transferred to the HPLC or the GC/MS. Were blanks done? Were calibrations performed (they are, but explain to the reader they are in 2.3)? I offer my thoughts, however useful they are, in the ‘Minor comments’ section below. I’d suggest a schematic too if possible (I don’t know if there is a figure limit in this journal).
3. Results: Overall this section is OK, however it can be improved. First, the discussion L148-151 is well known, and can be collapsed in a short sentence. Second, the authors always say “samples of a nicotine-free inhalator”. I thought only one inhaler was used in this study; for consistency, the authors need to address the “a” that makes the reader doubt how many there were. Third, I seem to bounce between numbers: 37 compounds (L98), 25 compounds with standards (L102), 13 ‘aroma’ compounds (Fig. 1), 12 compounds (Table 1), and 29 VOCs (Table 2). Something doesn’t add up, so can the authors clarify? It’s hard to follow the text in L75-77, and 97-101 and then compare to Tables 1 and 2. Can the authors clean the display of the data for the reader? I’d suggest a table with compound, detection method, and calibration type in Section 2.
4. Discussion: In Sect. 4.1, first paragraph, I struggle to understand how the results of other studies are used for discussion on the authors’ results. In the second paragraph, is there any inference on health concerning the finding that liquid in nicotine-free inhalers are similar to peppermint oils? What percentage of peaks were unidentified? Sect. 4.2 seems more appropriate for the introduction; I don’t see any results discussed or compared. Maybe take some compounds of this study that were emitted in high quantity and estimate a number of puffs necessary for them to be dangerous in the human body, for example? The same concern applies to Sect. 4.3. A link between the literature review and the findings of the study is necessary for it to be a discussion, otherwise the material is fit for a review article or the introduction section. Sect. 4 needs to put the results in context so that the reader can get a sense of whether the study finds nicotine-free inhalers are safe, unsafe, or more data is necessary.
5. Conclusions: Simple conclusions but given the analysis and results are bare, I am not sure I can say the results match the conclusion, especially with the 8-hr rule arbitrarily used by the authors. The English needs to be revisited as well.
Minor / Technical Comments:
L19: How many inhalers were used? Please check grammar (singular/plural consistency).
L34: Globally? Also, is a WHO URL an appropriate citation for a manuscript that is supposed to be peer reviwed? I ask that the authors find perhaps a citable report from the WHO and use that as a citation. Also, small detail, but “people/year” could be replaced with “people annually” to make it sound less colloquial.
L35: Similarly, if the reader clicks on the reference [2], the webpage itself cites citable articles. Do the authors not have access to the studies? I’d encourage the authors to bolster claims with actual studies rather than webpages, when possible.
L38: “The nicotine-free inhalator…” implies the authors are looking at one in particular. There is more than one nicotine-free inhalator (different manufacturers), so the sentence should reflect that. Or are the authors highlighting a specific one, perhaps of their own design?
L39-40: Redundant; either incorporate in previous sentence or please remove.
L44: If it does not require electronic operation, how do these work? Are there examples the authors can cite? Commercial or available in the literature?
L46: I’d perhaps suggest replacing “major cigarette components” with “typical cigarette / tobacco chemical markers”, or words to that effect.
L47: What are “natural essential oils and herbs”? Can the authors go deeper in the literature and give either class of compounds or compound names? The current phrasing comes across as pseudoscience and arguably doesn’t belong in the peer reviewed literature.
L48: Please fix the English “Until recently…impacts” sounds too colloquial.
L49: Volatile compounds, as in VOCs? Can the authors define VOC now or do they do it later in the text (Abstract doesn’t count)?
L51: Please define the verb ‘to vape’. The inexperienced reader might not understand vaping is an inhalation process. Is vaping specific to the aerosolization process of the liquid reservoir? Also, what are ‘safety and risks’ described here? Impact on human health? Is there any complication that results from vaping or e-cigarettes available in medical journals?
L56-58: I don’t understand this sentence. What does “analyzed with regard to the presence of a set of target components”? If it’s explained below, then briefly explain in L56 that this autosampler targets only specific chemical compounds. Also, is there a model of the Chemtekins, or is Chemtekins the company? Please refer to, e.g., Lee et al. 2018 doi: 10.3390/app8122699 or Dai et al. 2017 doi: 10.1016/j.microc.2017.02.029 in their descrpition
L60: Remove “machine”, and replace “(Chemtekins)” with “(make and model mentioned in the previous paragraph)” or words to that effect.
L62: Replace “and” with “resulting in”.
L64: Can the authors offer more description on these “basic procedures” that are recommended?
L65: Again, does this inhalator manufactured by Mihyang Med, Inc. have a name or model number? Also, the sentence confuses me. Are the authors describing what a nicotine-free inhaler is in this sentence? This should belong in the introduction (L43-44?). Did the authors mean “A nicotine-free inhaler (Model; Make; Country) was used for this study.” maybe?
L66-67: Is this specific for the Mihyang Med., Inc. model, or is it generally how all nicotine-free inhalers are made? Because if so, this belongs in the introduction.
L 69-70: What are Tenax TA and Sulficarb? Are they activated charcoal? A resin? What is DNPH? Why is DNPH a good sorbent for carbonyls? Typically, DNPH is used as a derivatizing agent for chromatography – the authors are encouraged to clarify that for readers that are unfamiliar with the technique. Why do the authors care about carbonyls – are carbonyls expected as replacements for PG, PEG, nicotine, nicotyrene etc.? Or are they expected to be common functional groups in oils used for vaping?
L70: I don’t think the acronym VOC has been defined yet (Abstract doesn’t count unless MDPI guidelines specify so).
L71-72: “The VOC concentration values and carbonyl compounds in vapor samples…” is misleading. Are both VOCs and the set of 10 carbonyls quantified, or just qualitatively identified? Or were only VOCs that are non-carbonyls quantified?
L72: What are “the two” products? I thought it was only one: the inhaler (is ‘inhalator’ a word?) made from Mihyang Med., Inc.?
L78-79: Again, model of the HPLC should be specified when possible.
L79-81: What does this mean? I don’t understand. Are the authors talking about a calibration? It sounds like it in the next sentences, but this sentence needs to be polished.
L86: Why was a 7:3 ratio of acetonitrile: water used?
L88: The authors already said they’d do a five-point calibration in L82; redundant.
L89: Fix “R2”, but more importantly, why do the authors choose not to display their calibrations in a figure? An example chromatogram at the very least? Coupled with the RSD, at least a table?
L94: Remove the parentheses; we know this from earlier.
L95: Remove “a”.
L97: Replace “quantitate” with “quantify”.
L97-101: ‘using’ standard materials? What does this mean? Are the authors saying that of the 37 compounds quantified, they had standards for calibrations for only 25 of them? Then I’d suggest rewording to: “were quantified relative to standards (Sigma-Aldrich…”. Then how have the remaining 12 been quantified? Direct the reader to Sect. 2.4.
L119: Replace “quantitated” with “quantified” (e.g., L121).
L122: Can the authors presenta formula or give an example for how they obtained a calibration based on carbon number theory and the predicted RF (area under the peak?)? Would RF be dependant on, say, the 70 eV electrons used in ionization?
L127-128: The first two sentences of Sect. 3 are redundant.
L129: I’d encourage the authors to be consistent: ‘volatile compounds’, ‘VOCs’, ‘volatile aroma compounds’… perhaps stick to the acronym, VOCs? Or, if ‘aroma’ means something specific, I’d encourage the authors to specify.
L132: Perhaps replace “major” with “most abundant”?
L146: Use a better word than “depicts”.
L151: “show a high volatility” is poor English, please rephrase.
L185-187: Redundant.
L189: Remove first sentence. On the second sentence, what is a ‘mint oil’? Also, this sentence would be more credible if we had a chromatogram and/or MS under each peak to verify.
L191: What is a ‘peppermint oil’ and why is it more stable?
L242: Citation needed.
L 276: “This is the first study to evaluate…” is improper English.
L283: Remove “In conclusion,”, and I would nuance “confirmed”.
Tables and Figures:
Figure 1: Why are there 13 compounds? I thought the ‘aroma’ compounds were 12?
Table 1: This table would make much more sense and the study would stand to more rigor had calibration curves, or example chromatograms, be included in the manuscript.
Table 2: Same as for Table 1.
References:
Dai et al. 2017 doi:10.1016/j.microc.2017.02.029 (already cited by the authors)
Lee, Y.S., Kim, K.H., Lee, S. S., Brown, R. J. C., Jo, S.H.; Analytical Method for Measurement of Tobacco-Specific Nitrosamines in E-Cigarette Liquid and Aerosol, Appl. Sci, 2018, 8, 2699; doi:10.3390/app8122699
Author Response
Reviewer 1
Summary:
This study is centered around a commercially available nicotine-free inhaler manufactured commercially by Mihyang Med, Inc. The authors use a commercially available e-cigarette autosampler to simulate puffs off the inhaler, and collect each puff on two filters: a filter with Tenax TA and Sulficarb sorbents for VOC analysis in GC/MS, and a filter with DNPH for the analysis of carbonyls in an HPLC. The study seems to be well thought through, however it is poorly written. Details of the setup are skipped, no chromatograms are available, and connection between the results and safe concentrations of inhaled compounds are not apparent. I have a few suggestions, offered below for the authors’ consideration, after which I think this manuscript will be of higher quality and appropriate for Applied Sciences. I do intend to raise an important query: the authors declare no conflicts of interest, yet to my understanding the study was carried out on an inhaler manufactured by Mihyang Med, Inc., who funded part of the study. Both the Abstract and the Conclusion sections claim there is no health risk if used<8h/day, but some compounds are above DNEL. This needs to be addressed clearly.
A) We have modified this sentence by following the valuable suggestion in page 1. We investigated the possibility that the nicotine-free inhalator should be safer than other smoking cessation aids (e.g., nicotine patch, drug: varenicline tartrate). The aroma non-smoking pipe is one of the preferable nicotine-free inhalator products in Korea. Hence, we conducted this study.
Major Comments:
[1] Abstract: No major comments; some grammatical errors but on the whole very concise and informative!
A) We have further modified as per suggestion.
[2] 1. Introduction: Overall, the introduction is very short and the English needs to be revisited. I am not sure if the authors do not have access to enough journals, but I would suggest a more in-depth review of the literature. How many types of these smoking-cessation devices that are not e-cigarettes or nicotine products exist (L35 the authors claim ‘many’, they can perhaps elaborate)? Are they mostly commercial or are there any being developed by research institutions? How does aerosolization work in these vs. e-cigarettes? Is the only difference the absence of nicotine, nicotyrene, PG, or PEG? Have there been studies that show these devices are effective? If most of this information is proprietary, perhaps the authors can clarify that.
A) We have modified this sentence in pages 1~2 to comply with your suggestion.
[3] 2. Materials and Methods: Sections 2.3 and 2.4 are fairly well described (however figures of calibrations or chromatograms are desired); but Sections 2.1 and 2.2 are too vague. It is my understanding that the nicotine-free inhaler (inhalator?) was connected to an autosampler, and the ~330 mL sample was sent to a filter, then the filter to the the HPLC. This isn’t clear, though. The authors need to go into more detail on their methods, be, and explain why they chose to detect carbonyls amongst other VOCs. Also, I do not understand how the sample on the filter gets transferred to the HPLC or the GC/MS. Were blanks done? Were calibrations performed (they are, but explain to the reader they are in 2.3)? I offer my thoughts, however useful they are, in the ‘Minor comments’ section below. I’d suggest a schematic too if possible (I don’t know if there is a figure limit in this journal).
A) We have added the importance of carbonyl compounds analysis in page 2. We added more explanation about the pretreatment methods for samples in section 2.2 and 2.3 (page 2~3).
[4] 3. Results: Overall this section is OK, however it can be improved. First, the discussion L148-151 is well known, and can be collapsed in a short sentence. Second, the authors always say “samples of a nicotine-free inhalator”. I thought only one inhaler was used in this study; for consistency, the authors need to address the “a” that makes the reader doubt how many there were. Third, I seem to bounce between numbers: 37 compounds (L98), 25 compounds with standards (L102), 13 ‘aroma’ compounds (Fig. 1), 12 compounds (Table 1), and 29 VOCs (Table 2). Something doesn’t add up, so can the authors clarify? It’s hard to follow the text in L75-77, and 97-101 and then compare to Tables 1 and 2. Can the authors clean the display of the data for the reader? I’d suggest a table with compound, detection method, and calibration type in Section 2.
A) We have modified as per suggestion (see pages 3~4). We summarized analytic information in Materials and Methods.
[5] 4. Discussion: In Sect. 4.1, first paragraph, I struggle to understand how the results of other studies are used for discussion on the authors’ results. In the second paragraph, is there any inference on health concerning the finding that liquid in nicotine-free inhalers are similar to peppermint oils? What percentage of peaks were unidentified? Sect. 4.2 seems more appropriate for the introduction; I don’t see any results discussed or compared. Maybe take some compounds of this study that were emitted in high quantity and estimate a number of puffs necessary for them to be dangerous in the human body, for example? The same concern applies to Sect. 4.3. A link between the literature review and the findings of the study is necessary for it to be a discussion, otherwise the material is fit for a review article or the introduction section. Sect. 4 needs to put the results in context so that the reader can get a sense of whether the study finds nicotine-free inhalers are safe, unsafe, or more data is necessary.
A) We have modified this sentence as per suggestion (pages 7~9).
[6] 5. Conclusions: Simple conclusions but given the analysis and results are bare, I am not sure I can say the results match the conclusion, especially with the 8-hr rule arbitrarily used by the authors. The English needs to be revisited as well.
A) We have modified this sentence as per suggestion (pages 1 and 9).
Minor / Technical Comments:
[7] L19: How many inhalers were used? Please check grammar (singular/plural consistency).
A) We have modified this sentence following your suggestion (page 1).
[8] L34: Globally? Also, is a WHO URL an appropriate citation for a manuscript that is supposed to be peer reviwed? I ask that the authors find perhaps a citable report from the WHO and use that as a citation. Also, small detail, but “people/year” could be replaced with “people annually” to make it sound less colloquial.
A) We have modified this sentence following your suggestion (page 1). However, we could not find out the latest report.
[9] L35: Similarly, if the reader clicks on the reference [2], the webpage itself cites citable articles. Do the authors not have access to the studies? I’d encourage the authors to bolster claims with actual studies rather than webpages, when possible.
A) We have modified this reference by following your suggestion.
[10] L38: “The nicotine-free inhalator…” implies the authors are looking at one in particular. There is more than one nicotine-free inhalator (different manufacturers), so the sentence should reflect that. Or are the authors highlighting a specific one, perhaps of their own design?
A) It is generally used with the name “nicotine free-inhalator” in paper and internet ad. All products of nicotine-free inhalator are similar in design.
[11] L39-40: Redundant; either incorporate in previous sentence or please remove.
A) We have deleted this sentence following your suggestion (page 1).
[12] L44: If it does not require electronic operation, how do these work? Are there examples the authors can cite? Commercial or available in the literature?
A) We have modified this sentence following your suggestion (page 2).
[13] L46: I’d perhaps suggest replacing “major cigarette components” with “typical cigarette / tobacco chemical markers”, or words to that effect.
A) We have modified this sentence following your suggestion (page 2).
[14] L47: What are “natural essential oils and herbs”? Can the authors go deeper in the literature and give either class of compounds or compound names? The current phrasing comes across as pseudoscience and arguably doesn’t belong in the peer reviewed literature.
A) We were not able to provide the exact ingredients because the information by the company was limited.
[15] L48: Please fix the English “Until recently…impacts” sounds too colloquial.
A) We have modified this sentence following your suggestion (page 2).
[16] L49: Volatile compounds, as in VOCs? Can the authors define VOC now or do they do it later in the text (Abstract doesn’t count)?
A) We have modified this sentence following your suggestion (page 2).
[17] L51: Please define the verb ‘to vape’. The inexperienced reader might not understand vaping is an inhalation process. Is vaping specific to the aerosolization process of the liquid reservoir? Also, what are ‘safety and risks’ described here? Impact on human health? Is there any complication that results from vaping or e-cigarettes available in medical journals?
A) We have modified this sentence following your suggestion (page 2).
[18] L56-58: I don’t understand this sentence. What does “analyzed with regard to the presence of a set of target components”? If it’s explained below, then briefly explain in L56 that this autosampler targets only specific chemical compounds. Also, is there a model of the Chemtekins, or is Chemtekins the company? Please refer to, e.g., Lee et al. 2018 doi: 10.3390/app8122699 or Dai et al. 2017 doi: 10.1016/j.microc.2017.02.029 in their description
A) We have modified this sentence following your suggestion (page 2).
[19] L60: Remove “machine”, and replace “(Chemtekins)” with “(make and model mentioned in the previous paragraph)” or words to that effect.
A) We have modified this sentence following your suggestion (page 2).
[20] L62: Replace “and” with “resulting in”.
A) We have modified this sentence following your suggestion (page 2).
[21] L64: Can the authors offer more description on these “basic procedures” that are recommended?
A) We already described all sampling procedures in the section 2.1 (page 2). We simply moved the sentences which describe the collection method of nicotine-free inhalator vapor to help reader’s understanding.
[22] L65: Again, does this inhalator manufactured by Mihyang Med, Inc. have a name or model number? Also, the sentence confuses me. Are the authors describing what a nicotine-free inhaler is in this sentence? This should belong in the introduction (L43-44?). Did the authors mean “A nicotine-free inhaler (Model; Make; Country) was used for this study.” maybe?
A) The nicotine-free inhalator manufactured by Mihyang Med, Inc. don’t have a model number. Product name is “Aroma non-smoking pipe”. This was used for this study.
[23] L66-67: Is this specific for the Mihyang Med., Inc. model, or is it generally how all nicotine-free inhalers are made? Because if so, this belongs in the introduction.
A) We have modified this sentence following your suggestion (page 2).
[24] L69-70: What are Tenax TA and Sulficarb? Are they activated charcoal? A resin? What is DNPH? Why is DNPH a good sorbent for carbonyls? Typically, DNPH is used as a derivatizing agent for chromatography – the authors are encouraged to clarify that for readers that are unfamiliar with the technique. Why do the authors care about carbonyls – are carbonyls expected as replacements for PG, PEG, nicotine, nicotyrene etc.? Or are they expected to be common functional groups in oils used for vaping?
A) DNPH cartridges are air-sampling media devices designated for sampling carbonyls (like formaldehyde) in ambient, indoor and industrial hygiene atmospheres. Carbonyls are trapped on a sampling cartridge containing high purity silica adsorbent coated with 2,4-dinitrophenylhydrazine (2,4-DNPH), where they are converted to the hydrazone derivatives. Tenax® TA is a traditional sorbent (porous polymer) for trapping medium to high boiling compounds; it is especially useful for low concentrations because of its low background. SulfiCarb is carbonised molecular sieves that are the strongest sorbents and are ideal for trapping the most volatile compounds.
[25] L70: I don’t think the acronym VOC has been defined yet (Abstract doesn’t count unless MDPI guidelines specify so).
A) We have modified this sentence following your suggestion (page 2).
[26] L71-72: “The VOC concentration values and carbonyl compounds in vapor samples…” is misleading. Are both VOCs and the set of 10 carbonyls quantified, or just qualitatively identified? Or were only VOCs that are non-carbonyls quantified?
A) We have modified this sentence following your suggestion (page 2).
[27] L72: What are “the two” products? I thought it was only one: the inhaler (is ‘inhalator’ a word?) made from Mihyang Med., Inc.?
A) We have modified this sentence following your suggestion (page 2).
[28] L78-79: Again, model of the HPLC should be specified when possible.
A) We have added this information following your suggestion (page 2).
[29] L79-81: What does this mean? I don’t understand. Are the authors talking about a calibration? It sounds like it in the next sentences, but this sentence needs to be polished.
A) We have modified this sentence following your suggestion (page 3).
[30] L86: Why was a 7:3 ratio of acetonitrile: water used?
A) We have added appropriate reference [15] in the sentence (page 3).
[31] L88: The authors already said they’d do a five-point calibration in L82; redundant.
A) We have modified this sentence following your suggestion (page 4).
[32] L89: Fix “R2”, but more importantly, why do the authors choose not to display their calibrations in a figure? An example chromatogram at the very least? Coupled with the RSD, at least a table?
A) We could not be included all information because analyzed compounds are too many.
[33] L94: Remove the parentheses; we know this from earlier.
A) We have modified this sentence following your suggestion (page 3).
[34] L95: Remove “a”.
A) We have modified this sentence following your suggestion (page 3).
[35] L97: Replace “quantitate” with “quantify”.
A) We have modified this sentence following your suggestion (page 3).
[36] L97-101: ‘using’ standard materials? What does this mean? Are the authors saying that of the 37 compounds quantified, they had standards for calibrations for only 25 of them? Then I’d suggest rewording to: “were quantified relative to standards (Sigma-Aldrich…”. Then how have the remaining 12 been quantified? Direct the reader to Sect. 2.4.
A) We have modified this sentence following your suggestion (page 3).
[37] L119: Replace “quantitated” with “quantified” (e.g., L121).
A) We have modified this sentence following your suggestion (page 3).
[38] L122: Can the authors present a formula or give an example for how they obtained a calibration based on carbon number theory and the predicted RF (area under the peak?)? Would RF be dependant on, say, the 70 eV electrons used in ionization?
A) We have added detailed description and Figure 1 to explain carbon number theory further (page 3). Response factor (RF) is not dependent on the operation mode (electron ionization mode at 70 eV) of MS system, but on sampling and analysis method (pretreatment, GC oven condition, etc.) applied for the experiment which can change RF value of target compounds to the MS system.
[39] L127-128: The first two sentences of Sect. 3 are redundant.
A) We have modified this sentence following your suggestion (page 4).
[40] L129: I’d encourage the authors to be consistent: ‘volatile compounds’, ‘VOCs’, ‘volatile aroma compounds’… perhaps stick to the acronym, VOCs? Or, if ‘aroma’ means something specific, I’d encourage the authors to specify.
A) We have modified this sentence following your suggestion (page 4).
[41] L132: Perhaps replace “major” with “most abundant”?
A) We have modified this sentence following your suggestion (page 4).
[42] L146: Use a better word than “depicts”.
A) We have modified this sentence following your suggestion (page 4).
[43] L151: “show a high volatility” is poor English, please rephrase.
A) We have modified this sentence following your suggestion (page 4).
[44] L185-187: Redundant.
A) We have modified this sentence following your suggestion (page 7).
[45] L189: Remove first sentence. On the second sentence, what is a ‘mint oil’? Also, this sentence would be more credible if we had a chromatogram and/or MS under each peak to verify.
A) We have modified this sentence following your suggestion. Peppermint is one type of mint. It was proved in ref 22 (page 7).
[46] L191: What is a ‘peppermint oil’ and why is it more stable?
A) We have modified this sentence following your suggestion (page 7).
[47] L242: Citation needed.
A) We have modified this sentence following your suggestion (page 8).
[48] L276: “This is the first study to evaluate…” is improper English.
A) We have modified this sentence following your suggestion (page 9).
[49] L283: Remove “In conclusion,”, and I would nuance “confirmed”.
A) We have modified this sentence following your suggestion (page 9).
Tables and Figures:
[50] Figure 1: Why are there 13 compounds? I thought the ‘aroma’ compounds were 12?
A) We have modified Figure 1 following your suggestion (page 5).
[51] Table 1: This table would make much more sense and the study would stand to more rigor had calibration curves, or example chromatograms, be included in the manuscript.
A) We have modified this sentence following your suggestion. We could not be included the analytic information because analyzed compounds are too many. And we summarized analytic information in Materials and Methods (page 5).
[52] Table 2: Same as for Table 1.
A) We have modified this sentence following your suggestion. We could not be included the analytic information because analyzed compounds are too many. And we summarized analytic information in Materials and Methods (page 6).
References:
[53] Dai et al. 2017 doi:10.1016/j.microc.2017.02.029 (already cited by the authors)
Lee, Y.S., Kim, K.H., Lee, S. S., Brown, R. J. C., Jo, S.H.; Analytical Method for Measurement of Tobacco-Specific Nitrosamines in E-Cigarette Liquid and Aerosol, Appl. Sci, 2018, 8, 2699; doi:10.3390/app8122699
A) We cannot figure out what reviewer wants to tell by this reference. If the reviewer still wants to point an issue, pls kindly describe.

Reviewer 2 Report
This is a well-written article, worthy of publishing. But I have concerns.
Abstract
1. In your abstract you say “To evaluate the reliability of cessation products,” That is really not what you are doing. Please correct this.
2. State an aim of the study in your abstract clearly and succinctly.
3. Why do you use the word “Additionally?” This seems to be the beginning of your results in your abstract. I would re-word.
4. Why in your abstract do you say less than 8 hours a day? How can you make that statement without backing it up? You may do so in the article itself, but you don’t in the abstract.
Introduction
1. I would delete the word good on line 35 page 1. Is there bad public policy?
2. I don’t think they are smoking cravings, but nicotine cravings. The action of smoking is a habit. There is a difference. (line 40)
3. In the US, there are nicotine inhalers. I would make the delineation clear that you are discussing only nicotine-free inhalers. In the US smokers use nicotine inhalers for smoking cessation as well as nicotine free inhalers. Make that more clear.
4. I would also make it clear that it seems to me you are evaluating the essential oils that people are using for the inhaler, as well as the volatile compounds used to make the inhaler work? Is that true? If so, that is unclear and must be resolved. If that is not true, then you must discuss all the compounds used in the inhalers and that combine with essential oils to make the inhalers work. Can you evaluate all the oils in that case?
Results: I don’t know what the derived no effect levels are and whether there is still danger from the chemicals even though there is no derived effect. The same is true for sensory irritation levels. Make it clearer in your tables if the DNEL, SI and OSHA levels are the recommended ones, and then your vapor samples compare to them.
I don’t think it is clear the way it is.
Table 1 has the volatile aroma compounds. I assume those are the essential oils?
And Table 2 has the volatile organic compounds. I assume that is what makes the oils vaporize? Make those more clear.
I would make it more clear in your tables that your last columns are the recommendations. And the levels you list for each chemical are those from one inhaler? Or from 4 puffs from one inhaler? Make that more clear as well.
I disagree with your findings about an 8-hour day. You need to make the reader aware of the levels you found (as you do), and clarify for the reader what the level you measured is (4 puffs? One complete inhaler? I don’t know). And then let the reader decide based on those facts whether or not the device is safe or not.
If I were the authors, I could compare the levels to those found in E-cigarettes or cigarettes and let the reader decide if they were safe or not.
I think you need to clarify this, and also list the guidelines from OSHA.
Author Response
Reviewer 2
Comments and Suggestions for Authors
This is a well-written article, worthy of publishing. But I have concerns.
Abstract
[54] 1. In your abstract you say “To evaluate the reliability of cessation products,” That is really not what you are doing. Please correct this.
A) We have modified this sentence following your suggestion (page 1).
[55] 2. State an aim of the study in your abstract clearly and succinctly.
A) We have modified this sentence following your suggestion (page 1).
[56] 3. Why do you use the word “Additionally?” This seems to be the beginning of your results in your abstract. I would re-word.
A) We have modified this sentence following your suggestion (page 1).
[57] 4. Why in your abstract do you say less than 8 hours a day? How can you make that statement without backing it up? You may do so in the article itself, but you don’t in the abstract.
A) We have modified this sentence following your suggestion (page 1).
Introduction
[58] 1. I would delete the word good on line 35 page 1. Is there bad public policy?
A) We have modified this sentence following your suggestion (page 1).
[59] 2. I don’t think they are smoking cravings, but nicotine cravings. The action of smoking is a habit. There is a difference. (line 40)
A) We have modified this sentence following your suggestion (page 2).
[60] 3. In the US, there are nicotine inhalers. I would make the delineation clear that you are discussing only nicotine-free inhalers. In the US smokers use nicotine inhalers for smoking cessation as well as nicotine free inhalers. Make that more clear.
A) We have modified this sentence following your suggestion (page 1).
[61] 4. I would also make it clear that it seems to me you are evaluating the essential oils that people are using for the inhaler, as well as the volatile compounds used to make the inhaler work? Is that true? If so, that is unclear and must be resolved. If that is not true, then you must discuss all the compounds used in the inhalers and that combine with essential oils to make the inhalers work. Can you evaluate all the oils in that case?
A) This study has been performed to evaluate the chemical compounds of vapor generated from nicotine-free inhalator. When inhaled, aroma oil naturally vaporized at room temperature. So We have modified introduction following your suggestion..
Results:
[62] I don’t know what the derived no effect levels are and whether there is still danger from the chemicals even though there is no derived effect. The same is true for sensory irritation levels. Make it clearer in your tables if the DNEL, SI and OSHA levels are the recommended ones, and then your vapor samples compare to them.
I don’t think it is clear the way it is.
A) Toxic guidelines (the derived no effect levels DNEL SI, OSHA) were already proven by paper and government institutes. Toxic guidelines were used to evaluate the working environment and indoor air.
[63] Table 1 has the volatile aroma compounds. I assume those are the essential oils?
A) It is assumed to be a volatile compounds vaporized from essential oil. And this compounds were contained in the peppermint oil.
[64] And Table 2 has the volatile organic compounds. I assume that is what makes the oils vaporize? Make those more clear.
A) We have modified a table caption following your suggestion (page 5~6).
[65] I would make it more clear in your tables that your last columns are the recommendations. And the levels you list for each chemical are those from one inhaler? Or from 4 puffs from one inhaler? Make that more clear as well.
A) We have modified a table caption following your suggestion (page 5~6).
[66] I disagree with your findings about an 8-hour day. You need to make the reader aware of the levels you found (as you do), and clarify for the reader what the level you measured is (4 puffs? One complete inhaler? I don’t know). And then let the reader decide based on those facts whether or not the device is safe or not.
If I were the authors, I could compare the levels to those found in E-cigarettes or cigarettes and let the reader decide if they were safe or not.
I think you need to clarify this, and also list the guidelines from OSHA.
A) We have modified this sentence following your suggestion (page 9).

Round 2
Reviewer 1 Report
Major comments:
The authors need to address the conflict of interest. Considering Mihyang Med, INC funded the study, I don’t understand how the authors declare no conflicts of interest.
I’m not convinved the conclusions support the results. The conclusion section could be strengthened by including some numbers, as well as discussing acrolein’s concentrations detected in high levels.
Minor comments:
Line 28: Acronym OSHA.
Line 66: Remove the clause in parentheses and use the next sentence to describe the specific brand/model used for this study.
Line 73: Acronym?
Line 152: This is simply a suggestion, but I wonder if comments like ‘excellent linearity’ are too subjective. The authors can consider instead just stating the correlation coefficient, and letting the reader decide whether it’s excellent or not.
Lines 214-216: How is this relevant to the study? Where’s the data? Example chromatograms (as I asked in the previous review)? As written, this sentence simply serves the purpose of positive advertisement for the manufacturer of the inhalator, which his not appropriate.
Line 233: Isn’t acrolein harmful to the human body? Can there me more discussion on the toxicity of acrolein?
Lines 267-269: Using only one citation to support such a claim is weak. I invite the authors to review the literature more in detail, assuming they have access to the required journals, to see if there are more studies on the antitumor properties of peppermint oil. As is, this comes across as pseudoscience. For that matter, the rest of the paragraph is based on this one reference, resulting in a fairly weak discussion.
Line 307: Please refrain from using improper language such as “…to quit smoking…”.
Author Response
Reply
21 May 2019
Decision: Accept after minor revision
Comments:
The conflict of interest question needs to be clarified. The authors declare no conflict of interest, but at the same time they acknowledge funding from the manufacturer of the nicotine-free inhalator, the device central to the study. It is not inherently wrong to accept such funding and publish the results, but the authors must be very specific to what extent they had some conflict of interest and how it was managed.
Decision Date
8 May 2019
Comments and Suggestions for Authors
Major comments:
The authors need to address the conflict of interest. Considering Mihyang Med, INC funded the study, I don’t understand how the authors declare no conflicts of interest.
ANS] We added Conflict of Interests and revised Acknowledgements.
I’m not convinved the conclusions support the results. The conclusion section could be strengthened by including some numbers, as well as discussing acrolein’s concentrations detected in high levels.
ANS] We have added the sentence in the discussion following your suggestion.Line293-298: Acrolein is considered to pose one of the non-cancerous health risk of all hazardous air pollutants [37]. In animal studies, decreased body weight gains were reported in rats, hamsters, monkeys, and rabbits exposed to 0.32-4.9 ppm acrolein. In the intermediate exposure study in rats (61 days, 24 hours/day), the NOAEL of 0.06 ppm and LOAEL of 0.32 ppm have been calculated [38]. The acrolein level (0.06 ppm) in the nicotine-free inhalator is the same as NOAEL for intermediate exposure. If daily exposure time is clearly shorter than 24 hours, it is considered safe.
Minor comments:
Line 28: Acronym OSHA.
ANS] We have modified this sentence following your suggestion.
Line28: the Occupational Safety and Health Administration → the Occupational Safety and Health Administration (OSHA)
Line 66: Remove the clause in parentheses and use the next sentence to describe the specific brand/model used for this study.
ANS] “Aroma non-smoking pipe” is product name. It is a single product in Mihyang Med Inc.
Line 73: Acronym?
ANS] We have modified this sentence following your suggestion.
Line73: the National Institute of Food and Drug Safety Evaluation → the National Institute of Food and Drug Safety Evaluation (NIFDS)
[(Ref. website= http://www.nifds.go.kr/en/index.do)]
Line 152: This is simply a suggestion, but I wonder if comments like ‘excellent linearity’ are too subjective. The authors can consider instead just stating the correlation coefficient, and letting the reader decide whether it’s excellent or not.
ANS] We have modified this sentence following your suggestion.
Line152: excellent linearity → linearity within acceptance criteria
[Ref.= Acceptance criteria of linearity: The linearity check should confirm that the method is linear e.g. the regression coefficient should be better than 0.99 over the working range and the working range should be fit for purpose (Guidance for the Validation of Analytical Methodology and Calibration of Equipment used for Testing of Illicit Drugs in Seized Materials and Biological Specimens, https://www.unodc.org/documents/scientific/validation_E.pdf)]
Lines 214-216: How is this relevant to the study? Where’s the data? Example chromatograms (as I asked in the previous review)? As written, this sentence simply serves the purpose of positive advertisement for the manufacturer of the inhalator, which his not appropriate.
ANS] We have modified this sentence following your suggestion.
Line 214-216: The composition of mint oils was found to be stable based on regular GC analysis. This finding suggests that mint oil, especially the more stable peppermint oils (e.g., relative to spearmint oil), is stable over time, so that the 12-month expiration date of mint oils could be extended [23].
Line 233: Isn’t acrolein harmful to the human body? Can there me more discussion on the toxicity of acrolein?
ANS] We have added the sentence (line293-298) in the discussion following your suggestion.
Lines 267-269: Using only one citation to support such a claim is weak. I invite the authors to review the literature more in detail, assuming they have access to the required journals, to see if there are more studies on the antitumor properties of peppermint oil. As is, this comes across as pseudoscience. For that matter, the rest of the paragraph is based on this one reference, resulting in a fairly weak discussion.
ANS] We have added two review articles following your suggestion.
1. Ali, B.; Al-Wabel, N. A.; Shams, S.; Ahamad, A.; Khan, S. A.; Anwar, F. Essential oils used in aromatherapy: A systemic review. Asian Pac. J. Trop. Biomed. 2015, 5, 601-611. https://doi.org/10.1016/j.apjtb.2015.05.007.
2. Balakrishnan, A. Therapeutic uses of peppermint-a review. J. Pharm. Sci. & Res. 2015, 7, 474-476.
Line 307: Please refrain from using improper language such as “…to quit smoking…”.
ANS] We have modified this sentence following your suggestion.
Line 307: to quit smoking → to stop smoking
Submission Date
22 March 2019
Date of this review
18 May 2019 01:08:23
Reviewer 2 Report
Thank you for making the changes I recommended.
Author Response
Replies to the reviewer’s comments
Thank you for making the changes I recommended.This article is acceptable now, however contains many major English language
flaws that need to be edited.
ANS] Yes, thanks per comments, it was edited once again!